# Impact of aerosol optics on vertical distribution of ozone in autumn over YRD

Shuqi Yan[2], Bin Zhu[1,*], Shuangshuang Shi[1], Wen Lu[1], Jinhui Gao[3], Hanqing Kang[1], Duanyang Liu[2]

[1]Collaborative Innovation Center on Forecast and Evaluation of Meteorological Disasters, Key Laboratory for Aerosol-Cloud-Precipitation of China Meteorological Administration, Key Laboratory of Meteorological Disaster, Ministry of Education (KLME), Special Test Field of National Integrated Meteorological Observation, Nanjing University of Information Science & Technology, Nanjing 210044, China
[2]Key Laboratory of Transportation Meteorology of China Meteorological Administration, Nanjing Joint Institute for Atmospheric Sciences, Nanjing 210041, China
[3]Plateau Atmosphere and Environment Key Laboratory of Sichuan Province, School of Atmospheric Sciences, Chengdu University of Information Technology, Chengdu 610225, China

*Correspondence to*: Bin Zhu (binzhu@nuist.edu.cn)

**Abstract.** Tropospheric ozone, an important secondary pollutant, is greatly impacted by aerosols within boundary layer (BL). Previous studies have mainly attributed ozone variation to either aerosol-BL or aerosol-photolysis interactions at near surface. In this study, we analyze the sensitivities of ozone response to aerosol mixing states (e.g., mixing behavior hypothesis of scattering and absorbing components) in the vertical direction and address the effects of aerosol-BL and aerosol-photolysis interactions on ozone profiles in autumn by WRF-Chem simulations. The aerosol internal mixing state experiment reasonably reproduces the vertical distribution and time variation of meteorological elements and ozone. Sensitive experiments show that aerosols lead to turbulent suppression, precursor accumulation, lower-level photolysis reduction and upper-level photolysis enhancement. Consequently, ozone basically decreases within entire BL during daytime (08:00~17:00), and the decrease is the least in external mixing state (2.0%) compared with internal (10.5%) and core-shell mixing states (8.6%). The photolysis enhancement is the most significant in external mixing state due to its strong scattering ability. By process analysis, lower-level ozone chemical loss is enhanced due to photolysis reduction and $NO_X$ accumulation under VOC-limited regime. Upper-level ozone chemical production is accelerated due to higher photolysis rate resulting from aerosol backscattering. Therefore, the increased ozone entrainment from aloft BL to surface induced by boosted ozone vertical gradient outweighs the decreased ozone entrainment induced by turbulent suppression after 11:00 am. Additional simulations support that aerosol effect on precursor, photolysis and ozone is consistent under different underlying surface and pollution conditions.

# 1 Introduction

Tropospheric ozone is an important secondary pollutant that is produced by the photochemistry of VOC (volatile organic

compounds) and $NO_x$. The variation of ozone is determined by the highly variable interactions among meteorology, precursors, photochemistry and aerosols. Tropospheric ozone, especially in the atmospheric boundary layer (BL), exerts side effects such as impairing human health, contributing to global warming and aggravating air pollution (Fu et al., 2019). Since 2013, the severe $PM_{2.5}$ pollution over East China has been mitigated but ozone concentration is increasing (Li et al., 2020). Therefore, the characteristic of ozone variation and its relationship with external factors need to be intensively studied.

The interactions between ozone and aerosols are complicated and have attracted wide concern in recent years. Aerosols can significantly affect ozone photochemistry by influencing photolysis process (herein called aerosol-photolysis interaction). The weakened solar radiation reaching the ground induced by aerosol extinction can decrease photolysis rate at the surface and within several hundred meters above the surface, thus inhibiting ozone production and resulting in lower ozone concentration (Gao et al., 2020; Jacobson, 1998; Li et al., 2011). Contrarily, scattering aerosols increase upward shortwave radiation which may promote ozone formation at a higher altitude (Gao et al., 2021a). Dickerson et al. (1997) and Shi et al. (2022) demonstrated that aerosol pollution can remarkably increase ultraviolet radiation at a few hundred meters above the aerosol layer, which accelerates photolysis and increase ozone concentration by about 3~20 ppb. Additionally, heterogeneous reactions on aerosol surface can also influence ozone chemistry (Jacob, 2000; Li et al., 2019; Lou et al., 2014).

Aerosols affect BL thermodynamics and ultimately result in ozone change, which has attracted much attention in recent years. The perturbation in radiation flux profile induced by aerosols can alter BL structure, thus influencing vertical mixing and affecting ozone and precursor concentration (herein called aerosol-BL interaction). Aerosols stabilize BL and suppress turbulent mixing (Ding et al., 2016; Li et al., 2017), which can inhibit the vertical exchange of ozone. Gao et al. (2018) studied the effect of black carbon (BC) on ozone variation within BL. BC weakens turbulent mixing and inhibits the higher ozone aloft being entrained downward. Additionally, the suppression of BL leads to the accumulation of $NO_x$ which promotes the formation of radicals and chemical production of ozone. The weakening in ozone mixing outweighs the enhancement in ozone chemical production, so the surface ozone is decreased during the daytime.

The effect of aerosols on BL is related to aerosol optics, which are determined by aerosol morphology (Liu et al., 2019), hygroscopicity (Zeng et al., 2019), coating process (Bond et al., 2006) and chemical composition. The aerosol chemical composition in East China is dominated by SNA (sulfate, nitrate and ammonium) (larger than 50%), followed by organic matter and BC (3~8%) (Yang et al., 2011; Tan et al., 2020, 2022). The contribution of SNA to total aerosol scattering coefficient can reach up to 60% (Tian et al., 2015), and BC accounts for more than 70% of total aerosol absorbing coefficient (Yang et al., 2008). Furthermore, aerosol optics are strongly affected by aerosol mixing states. Since the real-world mixing state is highly variable and hard to be explicitly resolved (Riemer & West, 2013), three typical mixing states are generally hypothesized by previous works: internal mixing, core-shell mixing and external mixing. The mixing state is largely affected by the mixing behavior of BC with other aerosol species. The freshly emitted BC is commonly externally mixed with other species, but it

will become more internally mixed due to coating process (Riemer et al., 2019). The BC light absorption can be amplified by a factor of 50~200% after being coating with scattering aerosols (Cappa et al., 2012; Jacobson, 2001; Liu et al., 2017). Accordingly, aerosol mixing state alters aerosol optical properties and affects its interactions with BL and photolysis. Gao et al. (2021b) found that aerosols result in smaller boundary layer height (PBLH) reduction in external mixing (11.6 m) than in core-shell mixing (24 m), consequently leading to different changes in photolysis rates and ozone concentration.

Many studies reveal the aerosol effect on ozone at near-surface level. Aerosols notably affect ozone photochemistry at all heights within BL and ultimately influence ozone vertical distribution and turbulent exchange. Therefore, the aerosol-induced ozone variation could have larger complexity and uncertainty in the vertical direction, which should be explored further. Additionally, previous studies explain ozone variation mainly by either aerosol-BL or aerosol-photolysis interaction, but relatively few of them consider these two mechanisms together. In this study, we aim to quantitatively reveal the impact of aerosols on ozone profile through the two pathways (aerosol-BL and aerosol-photolysis interactions) by WRF-Chem simulations, as well as how aerosol effect varies with aerosol mixing states in autumn season over the Yangtze River Delta Region (YRD), China. Heterogeneous chemistry is not included in this study. The manuscript is organized as follows. Section 2 introduces the data, model and sensitive experiments. Section 3.1 evaluates the model performances. Sections 3.2 to 3.4 reveal the characteristic of aerosol-BL and aerosol-photolysis interactions and their impacts on ozone variation. Section 4 discuss the robustness of simulation results under different conditions. Section 5 concludes the findings of this study.

# 2 Data, model and experiments

## 2.1 Data

A field campaign was conducted at an industrial zone in north Nanjing suburban (118.71 °E, 32.27 °N) from 15 October to 15 November 2020 (Figure 1). We collected the vertical profiles of meteorological elements (temperature, wind speed and direction) and air pollutants ($PM_{2.5}$, BC and ozone). Meteorological elements are measured by XLS-II tethered balloon system with a sounding balloon at 08:00 and 14:00 local time. The data are sampled each second until it loses signal. Air pollutants observation instruments are mounted on UAV platform. The UAV climbs vertically from the ground to about 1 km with a speed of 2m/s, and it descends along the same path at the same speed. The UAV is launched four times a day at around 09:00, 11:00, 14:00 and 16:00 (local time). The introduction of observation instruments of $PM_{2.5}$, BC and ozone can be referred to Shi et al. (2020, 2021). Meteorology and air pollutants profiles are averaged to 50 m intervals. These data are used to evaluate the model performance in the vertical direction.

The ground meteorology observation data is from MICAPS (Li et al., 2010), including temperature, wind speed and wind

direction that recorded every three hours. The ground air quality data is from China National Environmental Monitoring Center (https://www.aqistudy.cn/), including $PM_{2.5}$, ozone and other pollutants. We use the temperature, wind speed, wind direction, $PM_{2.5}$ and ozone data to evaluate the model performance on the time series of meteorological elements and air pollutants.

## 2.2    Model configuration

The model used in this study is the WRF-Chem (V3.9.1.1) model (Fast et al., 2006; Grell et al., 2005). It is the state-of-the-art atmospheric model that online couples meteorology and chemistry. Two domains are set up with the central point at the observation site (118.71 °E, 32.27 °N) (Figure 1). The parent domain has the size of 79×79 grids with the grid spacing of 27 km. The inner domain has the size of 79×79 grids with the grid spacing of 9 km, covering the most part of the Yangtze River Delta Region. To better describe the turbulent process, the vertical level is refined to 38 layers and 12 of which are below 2 km. All the model results are calculated at the nearest grid close to the observation site if not specified.

The anthropogenic emission inventory in the base year of 2020 is provided by MEIC from Tsinghua University (Zheng et al., 2018) (http://www.meicmodel.org/). MEIC includes major gaseous and aerosol species, e.g., $SO_2$, $NH_3$, VOCs, $NO_x$, BC, $PM_{2.5}$ and $PM_{10}$. The gas chemical mechanism is Carbon Bond Mechanism Z (CBMZ; Zaveri and Peters, 1999), and the aerosol chemical mechanism is Model for Simulating Aerosol Interactions and Chemistry with four bins (MOSAIC-4bin; Zaveri et al., 2008). These two chemical mechanisms are widely used for studying ozone chemistry. Detailed physical and chemical schemes are listed in Table 1.

The initial and boundary fields of meteorology are provided by ERA5 0.25°×0.25° reanalysis data (https://cds.climate.copernicus.eu/cdsapp#!/dataset/reanalysis-era5-pressure-levels?tab=form). The chemical initial and boundary fields are provided by WACCM (https://www2.acom.ucar.edu/gcm/waccm). They are all updated every 6 hours. The simulation starts at 08:00 on 15 October and ends at 20:00 on 15 November, and the first 72h is spin-up period. All the time here is local time (UTC+8).

## 2.3    Aerosol optics and sensitive experiments

In this work, the effect of aerosol optics on ozone profiles is addressed by its mixing states. We study three ideal types of mixing states: internal mixing, core-shell mixing and external mixing, which depend on the mixing behavior hypothesis of scattering and absorbing components. In internal mixing state, the relative fractions of chemical species in one particle are the same as that of the bulk aerosols. The complex refractive index (RI) of bulk aerosols is calculated by the vol-ume-averaged RI of all aerosol species, and then it is passed to Mie optical module to calculate the required optical parame-

ters (e.g., scattering coefficient, absorbing coefficient and single scattering albedo). The detailed formulas of aerosol optical parameters for MOSAIC sectional scheme are documented by previous works (e.g., Fast et al., 2006; Grell et al., 2005). In core-shell mixing, aerosol particles are hypothesized to be concentric spheres with BC as the core and non-BC aerosols as the coating shell (Riemer et al., 2019). The RI of the shell is the volume-averaged RI of non-BC aerosols, and the optics of core-shell mixed particles can also be treated by the Mie optical module (Ackerman & Toon, 1981). In external mixing state, each particle contains only one species with fixed optical characteristics. It is not included in the current WRF-Chem model, and the approximate treatment has been proposed by Gao et al. (2021b). In general, the Mie optical module separates BC aerosols from the bulk aerosols, and treats the optics of non-BC and BC aerosols individually.

To study the aerosol effect on ozone, four experiments are conducted (Table 2). The case "int" is the base experiment (the default option in WRF-Chem), in which the aerosols are internally mixed. The cases "csm" and "ext" are core-shell mixing and external mixing, respectively. The case "noARI" turns off aerosol-radiation feedback by setting aerosol optical depth as zero in radiation and photolysis modules. Therefore, the difference between noARI and three other experiments indicates the effect of aerosols in the corresponding mixing state.

One should note that the real-world aerosol mixing state varies with emission, meteorology, composition, and other factors. The dynamic evolution of aerosol mixing state and its influencing factors have not been addressed in most current 3D models (Matsui et al., 2013). This work addresses aerosol optics by the three ideal mixing states, which will inevitably cause the simulated aerosol optics deviating from observation.

# 3   Results

It is an obvious pollution stage on 2 November 2020. The model evaluation on profiles (Section 3.1) and the mechanism of aerosols affecting ozone variation (Sections 3.2 to 3.4) are presented at the Nanjing site during that day. The model evaluation on time series (Section 3.1) and the aerosol effect under different pollution conditions (Section 4) are presented during the simulation period (15 October to 15 November).

## 3.1   Model evaluations

Four additional sites around Nanjing, i.e., Changzhou (CZ), Huainan (HN), Maanshan (MS), and Huaian (HA) (Figure 1) are chosen to evaluate the performance on the time variation of meteorological parameters (temperature, wind speed and wind direction), $PM_{2.5}$ and ozone in the base experiment (internal mixing). The statistical metrics include index of agreement (IOA), mean bias (MB), root mean square error (RMSE), mean normalized bias (MNB) and mean fractional bias (MFB).

The calculations are from Lu et al. (1997), especially, the IOA of wind direction is from Kwok et al. (2010). Benchmark val-
ues of meteorology and air pollutants are derived from Emery et al. (2011) and EPA (2005; 2007). The temporal variations of
simulated meteorology and air pollutants are generally in good agreement with observations (Figure 2). From Table 3, tem-
perature presents the highest IOA, with a slightly large MB at HA site. The simulated wind direction is similar to observation,
and MB exceeds benchmark value at only one site. The simulated wind speed is a bit higher, which is because the WRF
model tends to overestimate wind speed due to the description of surface roughness (Jia and Zhang, 2020, 2021; Jiménez and
Dudhia, 2012). $PM_{2.5}$ is moderately overestimated, but all the metrics are within the benchmarks. The IOA of ozone exceeds
0.8 at all sites, and only one site shows a MNB out of benchmark. The model statistical metrics of $PM_{2.5}$ and ozone are con-
sistent with previous works (Chen et al., 2022; Hu et al., 2016; Singh et al., 2012; Zhang et al., 2014a). Generally, the base
experiment simulations on the temporal variation of meteorology and air pollutants are acceptable, which reasonably repro-
duces the observations in the atmosphere.
It is an obvious pollution stage on 2 November 2020 (Figure 2). We mainly evaluate the simulated profiles on that day. Fig-
ure 3 shows the model performance of meteorological parameters (temperature, wind speed and wind direction) and air pol-
lutants (ozone, $PM_{2.5}$ and BC). Seen from the profiles, temperature shows a similar pattern between simulation and observa-
tion, with the mean bias of 0.7 K and the maximum bias of 1.6K. The simulated wind direction and wind speed agree well
with observation, except that wind speed is overestimated for 1.2~1.9 m/s at 14:00. The ozone profile shows acceptable per-
formance, with the concentration being underestimated for about 2~12 ppb at 14:00 and 16:00. The simulated $PM_{2.5}$ profile
is generally consistent with observations. There is a moderate underestimation of 40~80 $\mu g/m^3$ at 11:00 below 800 m. BC
profile is almost close to observation, with the maximum bias of about 2~3 $\mu g/m^3$. Overall, the model reasonably captures
the vertical structure and temporal variation of meteorological elements, $PM_{2.5}$, BC and ozone, which is crucial for exploring
the mechanism of aerosol-BL and aerosol-photolysis interactions and explaining their impacts on ozone vertical profile.

## 3.2    Impact of aerosols on BL and $NO_x$

The effects of aerosols are detailly studied at the Nanjing site on 2 November 2020. Figure 4a shows the effect of aerosols on
PBLH. Aerosols consistently decrease PBLH in all mixing states, with the reduction of 152m (15.5%), 174m (17.8%) and
136m (14.0%) in internal, core-shell and external mixing conditions, respectively. External mixing exerts the weakest PBLH
reduction effect here, which is also reported by Gao et al. (2021b). The mechanism of BL suppression by aerosols has been
elucidated by many studies (e.g., Ding et al., 2016; Li et al., 2017). The suppression of BL can inhibit turbulent exchange
(Figure 4b) and favour the accumulation of precursor contents near the surface. $NO_x$ generally increases at all heights within
BL (Figure 4c), and this increase is significantly larger at lower heights than at upper heights. At near surface, the increase is
about 2 ppb for internal and core-shell mixing and about 1 ppb for external mixing.

The change in $NO_x$ may alter the ozone chemical regime and influence the sensitivity of ozone to VOC and $NO_x$. In this study, ozone chemical regime is indicated by $R=H_2O_2/HNO_3$. For Yangtze-River-Delta Region, ozone chemistry is in $NO_x$-limited regime if $R>0.8$ or in VOC-limited regime if $R<0.6$ or in transition regime if $0.6<R<0.8$ (Qu et al., 2021). The differences in R are small among various aerosol mixing states (Figure 5). Below the height of about 400m, ozone is $NO_x$-limited during 08:00~10:00 and VOC-limited after 10:00. While at the heights above 400m, ozone is dominantly VOC-limited in the whole daytime of 2 November. It indicates that despite the change in precursor concentrations, ozone chemical regime almost remains unchanged and it is mainly controlled by VOC. Therefore, the increase in $NO_x$ can enhance NO titration effect and inhibit ozone production, which will be further discussed in Section 3.4. Statistics on the entire model region also show that ozone chemical regime remains unchanged in most areas (>95%) and the dominant type is VOC-limited regime (>92%). Such is the case in the areas with urban or rural surfaces, and in the areas with high or low $NO_x$ emission rates.

## 3.3  Impact of aerosols on photolysis

The photolysis of $NO_2$ ($JNO_2$) and ozone (JO1D) are two major reactions that contribute to ozone production. In noARI condition, photolysis rates increase with height due to atmospheric extinction (figure not shown). When aerosol effect is included, photolysis rates decrease sharply at lower level but increase at upper level in all mixing states (Figure 6a and b). At the surface level, the relative change of $JNO_2$ and JO1D in internal mixing state is approximately -30%, which is similar to the value of -22.6% reported by Wu et al. (2020) and -23.0% by Zhao et al. (2021) that conducted in autumn and winter seasons. Notably, in external mixing state, the lower-level decrease is the smallest and the upper-level increase is the largest, with the maximum increase of about 10%. Also, the height where photolysis rate (e.g., $JNO_2$) starts to increase is lower in external mixing state (~700m) than in other mixing states (~1200m).

The significant differences in photolysis change can be explained by aerosol optical properties and its impact on radiation transfer. The aerosol extinction coefficient shows no obvious differences under the three mixing states, with the maximum difference of about 0.05 km$^{-1}$ (Figure 6c). However, the single scatter albedo (SSA) shows distinct differences (Figure 6d). SSA is about 0.8~0.9 in internal and core-shell mixing conditions below 2000m, and it is about 0.90~0.98 in external mixing condition which indicates a strong scattering ability. Zeng et al. (2019) also found that SSA is the largest in external mixing state compared with other mixing states. Therefore, it will backscatter more solar radiation to the upper level (Figure 6e) and promotes photolysis there (Figure 6a and b). Shi et al. (2022) have provided observational evidence that aerosols can increase upwelling shortwave radiation and promote photolysis at the upper level.

## 3.4 Impact of aerosols on ozone profile

Figure 7 shows the ozone profile in various mixing states. We focus on the ozone within BL in the daytime. During 08:00~11:00, the BL is in increasing stage, and ozone increases with height within BL. The average changes in ozone under internal, core-shell and external mixing are -9.7 ppb (-15.8%), -8.5 ppb (-13.8%) and -3.3 ppb (-5.4%), respectively. As BL develops during 11:00~17:00, ozone shows a strong positive gradient near the surface, uniform distribution above the surface and negative gradient at upper BL. The average change in ozone under internal, core-shell and external mixing is -7.3 ppb (-9.3%), -5.9 ppb (-7.5%) and -1.0 ppb (-1.2%), respectively. During the daytime (08:00~17:00), ozone reduction is larger in internal (10.5%) and core-shell mixing states (8.6%) and the smallest in external mixing state (2.0%). The reduction (about 3~13%) is the largest at near surface, which is due to that the $NO_x$ accumulation and photolysis inhibition are more profound at near surface. Other studies also reveal that ozone reductions caused by aerosols are approximately in the range of 10~20% (e.g., Gao et al., 2020; Qu et al., 2021; Yang et al., 2022). Above surface where the layer is more well-mixed, ozone reduction is relatively weaker. It can be inferred that diurnal ozone concentration is generally reduced in all mixing states and at all heights within BL. The reduction is the smallest in external mixing state. It could be because the enhanced NO titration effect associated with $NO_x$ accumulation is weaker in external mixing than in other mixing states (Figure 4c). Also, externally mixed aerosols lead to less photolysis suppression in the lower level and larger photolysis enhancement in the upper level (Figure 6a and b), which will partly counteract the reduction in ozone concentration.

To illustrate the mechanism of aerosols affecting ozone variation, we perform process analysis on ozone (Zhang et al., 2014b). In this study, ozone is decomposed into vertical mixing (VMIX), net chemical production (CHEM) and advection (ADVC; including horizontal and vertical advection) (Figure 8). The sign of CHEM depends on the competition between ozone production and loss. Under the effect of aerosols, CHEM shows negative change at near surface and positive change from lower to upper BL (Figure 8f-h). The negative CHEM change can be explained by the decrease in photolysis rate (Figure 6a and b) and the increase in NO titration associated with $NO_x$ accumulation (Figure 4c). Photolysis reduction may inhibit ozone production, and the increased NO titration consumes more ozone under VOC-limited regime (Figure 5f). From lower to upper BL, the positive CHEM change is dominantly contributed by the significant photolysis enhancement (Figure 6a and b). Since photolysis enhancement is the strongest in external mixing state, the increase in CHEM is the largest compared with other mixing states (Figure 8f-h). Above BL, especially between the solid and dash lines, the change in CHEM is negative due to the inhibited turbulent transport of $NO_x$ from the BL.

The variation in ozone photochemistry indicated by CHEM can influence VMIX which depends on ozone vertical gradient and turbulent exchange. In noARI condition, VMIX presents three distinct entrainment zones according to its signs: positive zone near the surface, negative zone at lower-to-middle BL, and time-variant zone at upper BL (near PBLH). VMIX is positive near the surface and negative at lower-to-middle BL (Figure 8a), because the higher concentration of ozone aloft is en-

trained downward by turbulent mixing. The time-variant VMIX zone at upper BL, specifically, negative values during 08:00~11:00 and positive values during 11:00~16:00 (Figure 8a), is determined by the relationship between PBLH diurnal variation and ozone vertical gradient below PBLH. During 08:00~11:00, ozone gradient at upper BL is positive (Figure 7a), which causes entrainment loss at that height. Above BL where ozone gradient and turbulent mixing are weak, ozone vertical exchange is not significant. Consequently, VMIX is negative at upper BL. During 11:00~16:00, ozone gradient at upper BL is negative (Figure 7b), which causes entrainment gain at that height and the positive VMIX at upper BL. Under the effect of aerosols, VMIX notably increases near the surface and basically decreases above surface in all mixing states especially after 11:00 (Figure 8b-d). It is because that the reinforced NO titration effect near surface and the enhanced photolysis aloft strengthen the ozone vertical gradient. The increase in gradient promotes ozone vertical exchange, compensating for the weakened ozone entrainment due to turbulent suppression, and instead, more ozone aloft are entrained to near surface (Gao et al., 2020, 2021a). At upper BL, the change in VMIX is negative during 08:00~11:00 and positive during 11:00~16:00. It is possibly due to that the negative and positive VMIX zones in Figure 8a move downward as PBLH decreases. The contribution of ADVC is relatively not important compared with VMIX and CHEM.

# 4   Discussions

Above we have presented the variation in photolysis rates, ozone precursors and ozone concentration induced by aerosols in a polluted day. To make the results more convincing, we perform additional analysis and simulations. The effect of aerosols on ozone may depend on locations and underlying surface type, e.g., urban and rural surfaces (Zhu et al., 2015). From Table 4, the qualitative results are consistent among different sites and underlying surfaces. Ozone shows a consistent decreasing and $NO_x$ shows a consistent increasing feature under the effect of aerosols. Photolysis rate (e.g., $JNO_2$) basically presents the dual change (i.e., lower-level decreasing and upper-level increasing). Comparing the three mixing types, the changes in photolysis rates, ozone precursors and ozone concentration caused by externally mixed aerosols are most favourable for mitigating ozone reduction. The mechanisms have been explained in previous sections.

The ozone variations during representative clean and polluted episodes are shown in Table 5. The ozone concentrations within BL in internal mixing experiment are consistently reduced during all episodes. The core-shell mixing state shows slightly lower reductions than internal mixing, and the ozone reductions are the least in external mixing state. The differences in ozone relative changes between clean and polluted episodes are distinct. For example, in the internal mixing state, the relative reductions are about 0~5% in clean episodes and 6~11% in polluted episodes, indicating that the aerosol effect is more profound under high aerosol contents. On 2 November which is the highest pollution episode during the study period, the relative changes of ozone are approximately -11~-2% in three mixing states. It can be inferred that aerosol effect on photolysis rates, ozone precursors and ozone concentration might be consistent under different underlying surface and pollution

conditions, and it is more significant in high aerosol conditions.

# 5  Conclusions

Previous studies mainly focus on the relationship between aerosols and ozone at near surface and attribute ozone variation to either aerosol-BL or aerosol-photolysis interactions. In this work, we explore the sensitivities of ozone response to aerosol mixing states in the vertical direction by WRF-Chem simulations from 15 October to 15 November 2020 over the Yangtze River Delta Region. Generally, the model reasonably captures the vertical profiles and temporal variation of meteorological elements, ozone, $PM_{2.5}$ and BC. Sensitive experiments show that:

Aerosols influence ozone vertical variation through aerosol-BL and aerosol-photolysis interactions. Aerosol inhibits BL development, resulting in more $NO_x$ accumulated within BL and a stronger NO titration effect under VOC limited regime. The PBLH reduction and $NO_x$ accumulation are the smallest in external mixing state. Despite the change in precursor concentration, ozone chemical regime is still dominantly controlled by VOC (>95%) under different underlying surface and emission conditions. Aerosols inhibit photolysis at lower level (~-30%) but enhance photolysis at upper level (~10%) due to aerosol backscattering. The enhanced photolysis is more obvious in external mixing state owing to its strong scattering ability.

Aerosols basically lead to ozone reduction (2~10%) at all heights within BL during the daytime (08:00~17:00), with the least reduction (2.0%) in external mixing state. Such ozone variation is attributed to the changes in VMIX, CHEM and ADVC. CHEM decreases at near surface due to photolysis reduction and $NO_x$ accumulation, but increases from lower to upper BL due to photolysis enhancement. The photolysis reduction and $NO_x$ accumulation at lower level lead to ozone depletion and stronger vertical gradient, which promotes higher concentration of ozone aloft being entrained downward. Therefore, VMIX increases at near surface but decreases at lower-to-middle BL. VMIX variation at upper BL (near PBLH) is complex, which is determined by the relationship between PBLH diurnal variation and ozone gradient near PBLH. Additional analysis indicate that aerosols could consistently cause precursor accumulation, dual change of photolysis and ozone reduction under different underlying surface and pollution conditions.

*Code and data availability*. Some of the data repositories have been listed in Section 2. The other data, model outputs and codes can be accessed by contacting Bin Zhu via binzhu@nuist.edu.cn.

*Author contributions*. SY performed the model simulation, data analysis and manuscript writing. BZ proposed the idea, supervised this work and revised the manuscript. SS provided the data at observation site. WL, JG and HK offered helps to the

model simulation. DL helped the revision of the manuscript.
*Competing interests.* The authors declare that they have no conflict of interest.
*Acknowledgements.* This work is supported by the National Natural Science Foundation of China (Grant Nos. 92044302,
42192512 and 42275115).

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

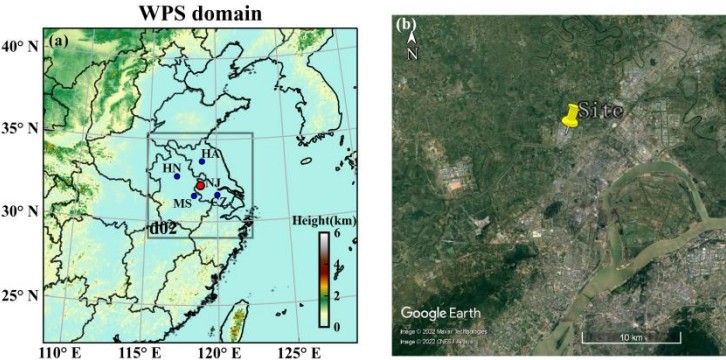

Figure 1.   The model simulation domain (a) and the surrounding area of the observation site (b). The red point in (a) and the yellow symbol in (b) are the observation site in Nanjing (NJ). The four blue points in (a) are Changzhou (CZ), Huainan (HN), Maanshan (MS) and Huaian (HA) sites.


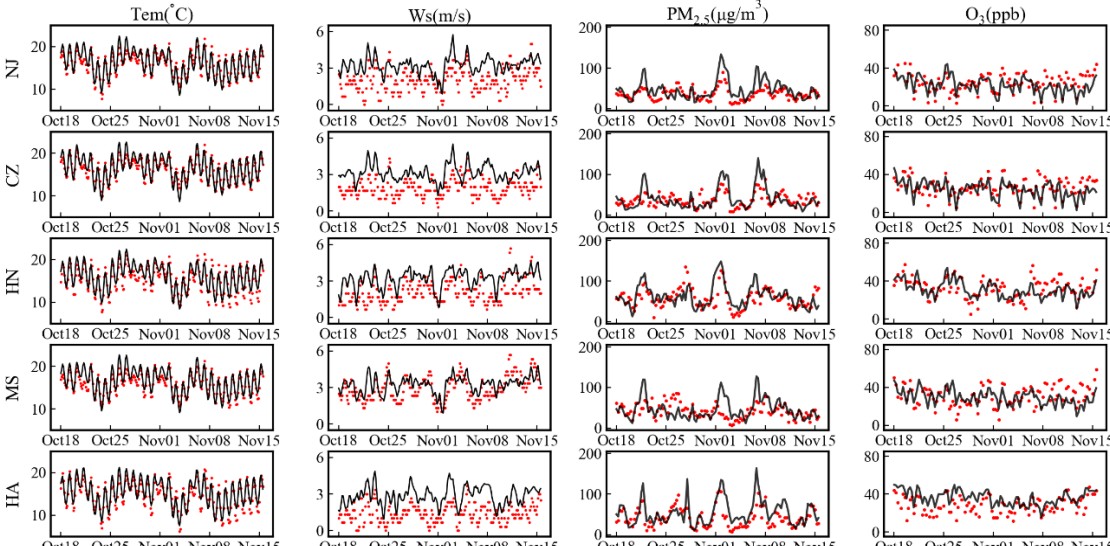

Figure 2.    Model evaluations on the time series on temperature (Tem), wind speed (Ws), PM$_{2.5}$ and Ozone at five sites.
The Changzhou (CZ), Huainan (HN), Maanshan (MS) and Huaian (HA) sites are located to the east, west, south and
north of Nanjing, respectively. The red dots are observations and black lines are simulations (after 3-point running av-
erage). The time range is from 08:00 on 15 October to 20:00 on 15 November.

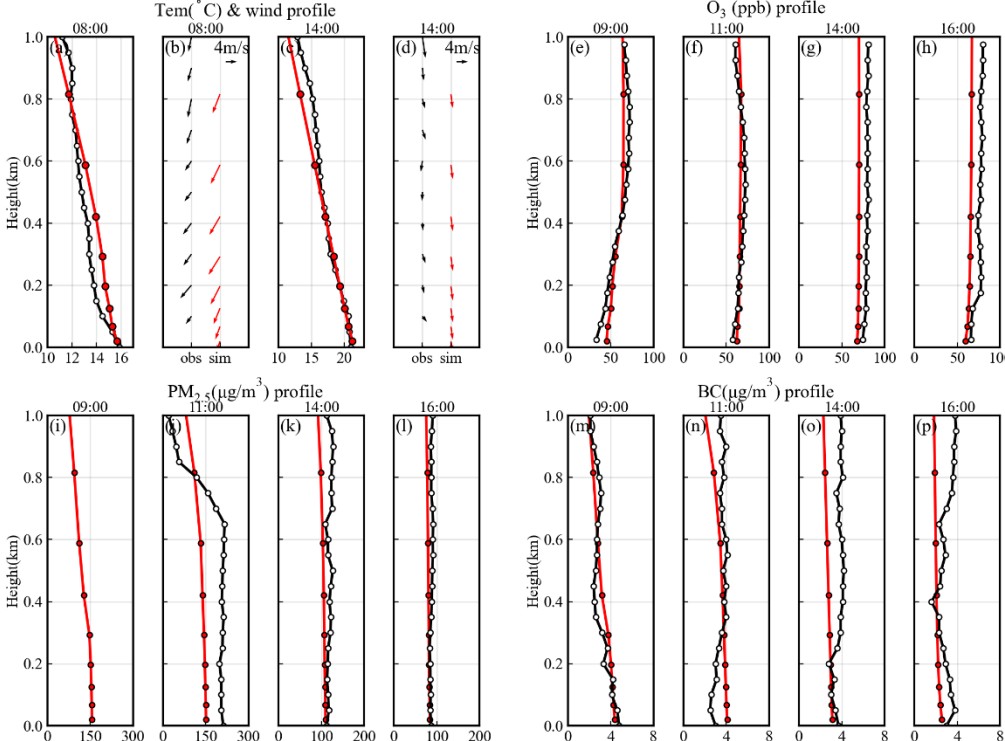


Figure 3. Model evaluations on the profiles of temperature, wind (vectors), ozone, $PM_{2.5}$ and BC on 2 November 2020. The black color is observation and the red color is simulation. The $PM_{2.5}$ observation data at 09:00 is missing due to instrument failure.


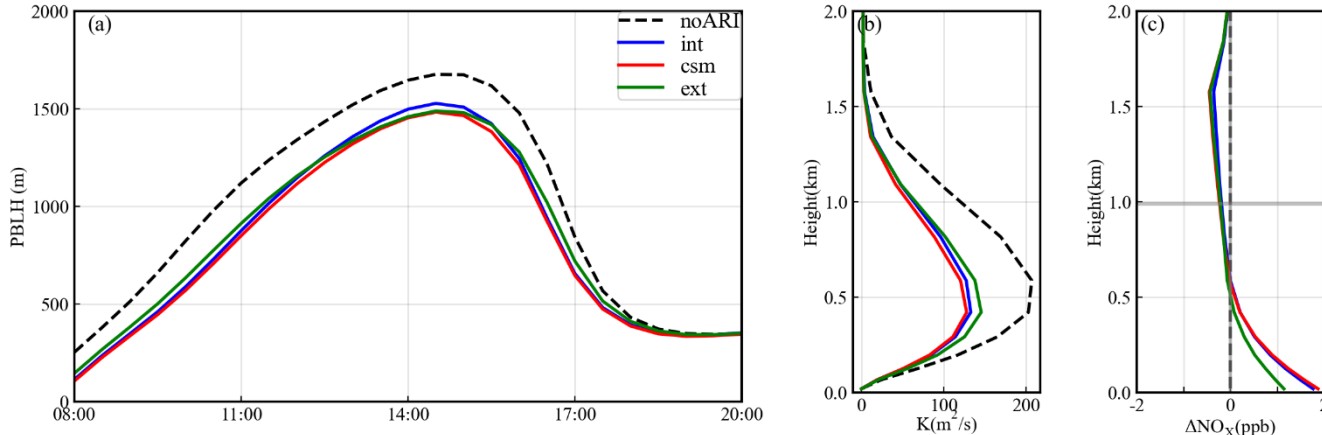


Figure 4. Time series of PBLH (a), profile of turbulent exchange coefficient K (b) and aerosol-induced change of
$NO_x$ profile (c) under different mixing states. The horizontal line in (c) is the PBLH of the base experiment. The pro-
files and PBLH in (c) are averaged during 08:00~17:00.

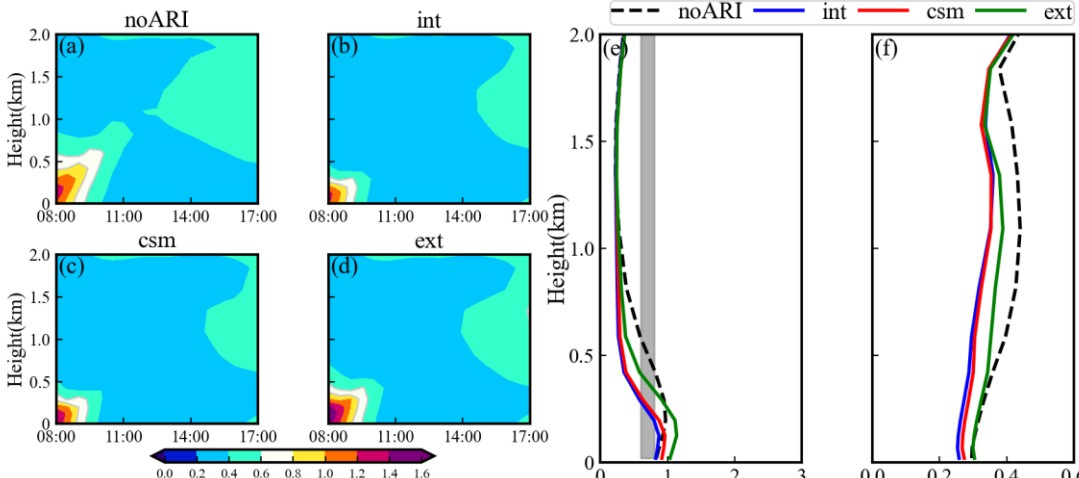


Figure 5.   (a-d) Time-height distribution of ozone chemical regime (indicated by R=$H_2O_2$/$HNO_3$) in different aerosol mixing states. (e-f) Profiles of R averaged during 08:00~10:00 and 10:00~17:00, respectively. The white contours in (a-d) and the grey strips in (e-f) represent the transition regime (0.6<R<0.8).

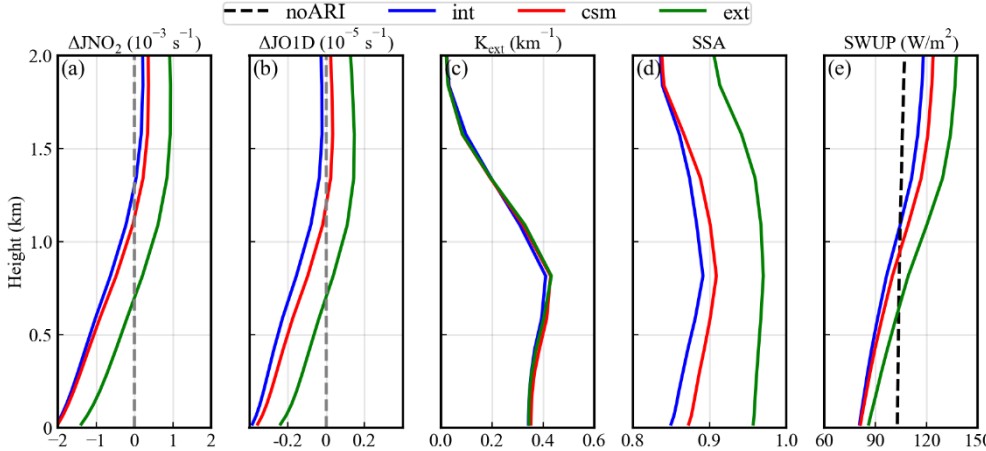

458

Figure 6. Comparisons of JNO$_2$ (a), JO1D (b), aerosol extinction coefficient (c), single scatter albedo (d) and upwelling shortwave flux (e) profiles among different mixing states. For JNO$_2$ and JO1D, the profiles are the changes with respect to noARI condidition. Profiles are time averages during 11:00~17:00.

463

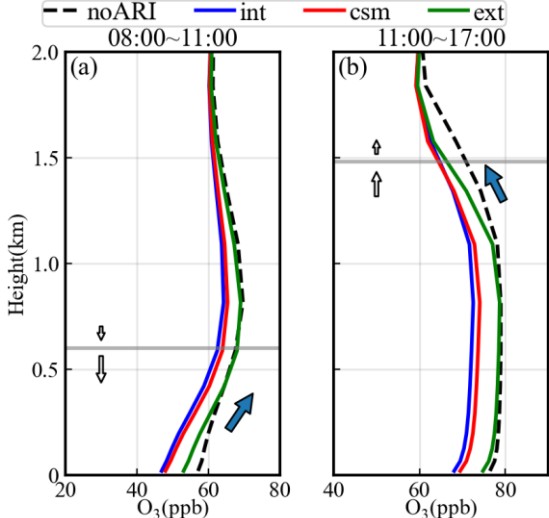

464

Figure 7. Ozone profiles under different mixing states. (a) 08:00~11:00 average. (b) 11:00~17:00 average. The horizontal line is PBLH. The blue arrows highlight the ozone vertical gradient at corresponding heights. The white arrows qualitatively describe the direction and magnitude of ozone turbulent exchange at the corresponding heights above or below PBLH.


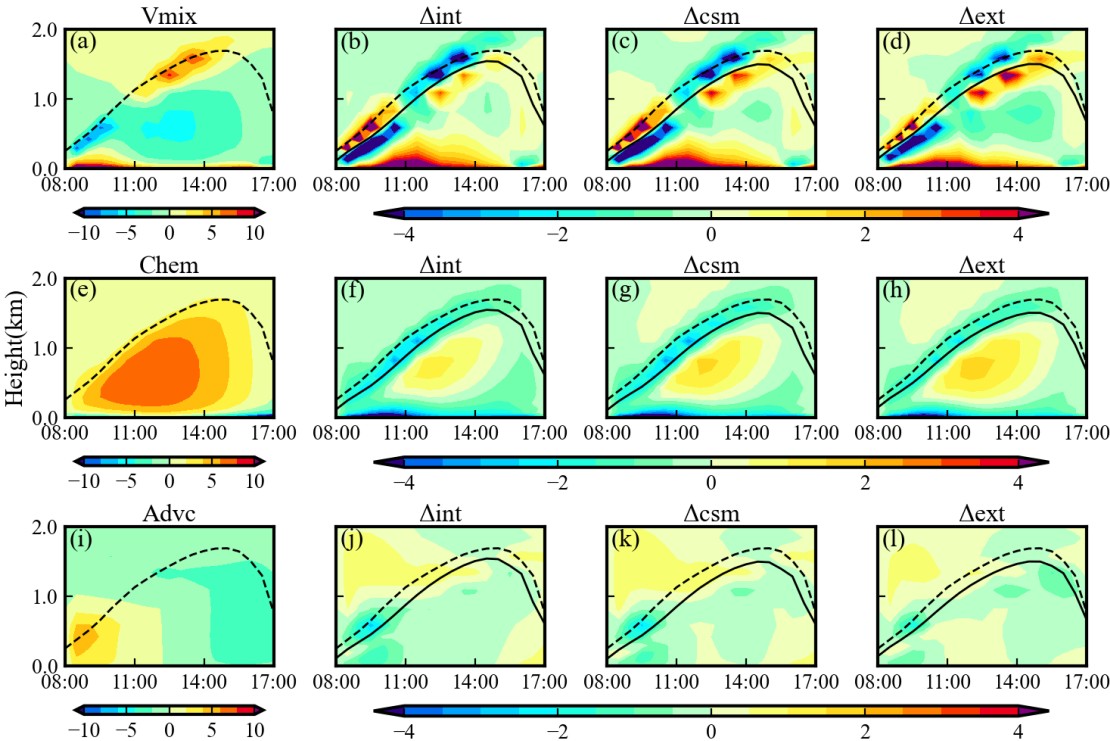


Figure 8. The time-height distribution of process tendencies (ppb/h) that contribute to ozone variation. The three rows are Vmix, Chem and Advc, respectively. The first column is the ozone tendency in noARI condition, and the rest three columns are the changes in ozone tendency under different aerosol mixing states.


Table 1.    Physical and chemical parameterization schemes.

| Scheme | Option |
| --- | --- |
| Boundary layer | YSU |
| Microphysics | Lin |
| Longwave radiation | RRTMG |
| Shortwave radiation | RRTMG |
| Land surface | Noah |
| Grid nudging | On |
| Observation nudging | Off |
| Gas phase chemistry | CBMZ |
| Aerosol chemistry | MOSAIC-4bin |
| Aerosol-radiation feedback | On |
| Aerosol optical properties | Varies with experiments |


Table 2.   Settings of sensitive experiments.

| Case name | Aerosol mixing states |
|---|---|
| int | internally mixed; base experiment |
| csm | core-shell mixed |
| ext | externally mixed |
| noARI | turn off aerosol-radiation feedback |
| Effect | Description |
| $\Delta$int=int-noARI | effect by internal mixing |
| $\Delta$csm=csm-noARI | effect by core-shell mixing |
| $\Delta$ext=ext-noARI | effect by external mixing |


Table 3.    The statistic metrics of the model performance on time series of temperature (Tem), wind speed (WS), wind
direction (WD), PM$_{2.5}$ and ozone. The benchmark values are from Emery et al. (2011) and EPA (2005; 2007). Metrics
that out of benchmarks are marked with red. (Nanjing:NJ, Changzhou:CZ, Huainan:HN, Maanshan:MS, Huaian:HA)

| Variable | Metric | NJ | CZ | HN | MS | HA | benchmark |
|---|---|---|---|---|---|---|---|
| | IOA | 0.97 | 0.97 | 0.96 | 0.97 | 0.96 | >0.8 |
| Tem | MB | 0.18 | 0.18 | 0.42 | 0.31 | 0.50 | <±0.5 |
| | RMSE | 1.07 | 1.07 | 1.43 | 1.10 | 1.52 | |
| | IOA | 0.64 | 0.63 | 0.66 | 0.71 | 0.64 | >0.6 |
| WS | MB | 0.47 | 0.68 | 0.52 | -0.05 | 0.71 | <±0.5 |
| | RMSE | 1.13 | 1.06 | 1.09 | 0.88 | 1.09 | <2 |
| | IOA | 0.94 | 0.93 | 0.93 | 0.95 | 0.88 | |
| WD | MB | -3.32 | 10.47 | 9.91 | -4.65 | 6.16 | <±10 |
| | RMSE | 35.91 | 38.53 | 46.31 | 36.56 | 52.92 | |
| | IOA | 0.74 | 0.84 | 0.83 | 0.64 | 0.86 | |
| PM$_{2.5}$ | MNB | 0.26 | 0.01 | 0.12 | 0.36 | 0.34 | |
| | MFB | 0.17 | -0.04 | 0.06 | 0.23 | 0.22 | <±0.6 |
| | IOA | 0.87 | 0.88 | 0.91 | 0.83 | 0.88 | |
| Ozone | MNB | -0.07 | -0.03 | 0.03 | 0.03 | 0.20 | <±0.15 |
| | MFB | -0.15 | -0.07 | 0.02 | 0.03 | 0.17 | |


Table 4. The diurnal averaged (08:00~17:00) variations of ozone, $NO_x$ and $JNO_2$ variations caused by different aerosol mixing states at four sites around Nanjing, urban areas and rural areas. The urban (rural) means the averages over urban (rural) surfaces of the model grids. The date is 2 November 2020. (Changzhou:CZ, Huainan:HN, Maanshan:MS, Huaian:HA)

| | CZ | HN | MS | HA | urban | rural |
|---|---|---|---|---|---|---|
| $\Delta$Ozone (ppb) (0.0~1.5km) | | | | | | |
| $\Delta$int | -8.8(-12.1%) | -3.5(-5.8%) | -6.0(-8.3%) | -5.8(-8.7%) | -5.3(-8.5%) | -5.0(-7.9%) |
| $\Delta$csm | -8.1(-11.1%) | -3.1(-5.1%) | -4.8(-6.7%) | -5.0(-7.4%) | -4.4(-7.1%) | -4.2(-6.7%) |
| $\Delta$ext | -3.7(-5.1%) | -1.2(-2.0%) | -0.8(-1.0%) | -1.4(-2.1%) | -1.1(-1.8%) | -0.9(-1.5%) |
| $\Delta NO_x$ (ppb) (0.0~1.5km) | | | | | | |
| $\Delta$int | 0.7(+16.2%) | 0.6(+20.7%) | 0.6(+12.3%) | 0.4(+20.6%) | 0.7(+11.4%) | 0.5(+16.2%) |
| $\Delta$csm | 0.7(+16.8%) | 0.7(+22.3%) | 0.5(+11.3%) | 0.4(+20.9%) | 0.7(+10.3%) | 0.5(+15.0%) |
| $\Delta$ext | 0.6(+14.3%) | 0.5(+15.5%) | 0.2(+3.4%) | 0.2(+10.2%) | 0.3(+5.0%) | 0.2(+6.3%) |
| $\Delta JNO_2$ ($10^{-3}s^{-1}$) (0.0~1.0km) | | | | | | |
| $\Delta$int | -1.4(-25.7%) | -1.0(-16.1%) | -1.5(-23.1%) | -1.3(-21.0%) | -0.9(-18.7%) | -1.1(-18.6%) |
| $\Delta$csm | -1.4(-24.6%) | -1.0(-15.5%) | -1.4(-21.9%) | -1.3(-20.4%) | -0.8(-17.3%) | -1.0(-17.4%) |
| $\Delta$ext | -0.9(-15.9%) | -0.5(-7.1%) | -0.7(-11.1%) | -0.7(-10.6%) | -0.4(-7.6%) | -0.4(-7.3%) |
| $\Delta JNO_2$ ($10^{-3}s^{-1}$) (1.0~1.5km) | | | | | | |
| $\Delta$int | -0.0(-0.5%) | -0.0(-0.2%) | -0.1(-1.7%) | -0.1(-1.8%) | -0.2(-3.5%) | -0.1(-1.4%) |
| $\Delta$csm | 0.2(+2.3%) | 0.0(+0.5%) | 0.0(+0.7%) | -0.0(-0.5%) | -0.1(-1.7%) | 0.0(+0.4%) |
| $\Delta$ext | 0.7(+9.4%) | 0.5(+7.5%) | 0.7(+9.7%) | 0.6(+8.2%) | 0.4(+6.4%) | 0.7(+9.3%) |

Table 5. The diurnal averaged (08:00~17:00) quantities within BL during some representative clean and polluted episodes. The $PM_{2.5}$ ($\mu g/m^3$) and ozone (ppb) are the values in the internal mixing state. The last three columns are the changes and relative changes of ozone under different mixing states.

| Date | $PM_{2.5}$ | Ozone | $\Delta$int | $\Delta$csm | $\Delta$ext |
|---|---|---|---|---|---|
| Clean episode | | | | | |
| 10-19 | 32 | 53 | -1.9 (-3.5%) | -1.7 (-3.1%) | +0.0 (+0.0%) |
| 10-20 | 18 | 49 | -0.8 (-1.5%) | -0.7 (-1.4%) | +0.1 (+0.1%) |
| 10-25 | 28 | 53 | -2.0 (-3.6%) | -1.9 (-3.5%) | -0.2 (-0.3%) |
| 11-03 | 33 | 39 | -0.7 (-1.8%) | -0.7 (-1.8%) | -0.4 (-1.1%) |
| 11-05 | 17 | 44 | -1.5 (-3.3%) | -1.5 (-3.3%) | -0.9 (-1.9%) |
| 11-12 | 23 | 36 | -1.9 (-4.9%) | -1.9 (-4.9%) | -0.9 (-2.5%) |
| Polluted episode | | | | | |
| 10-22 | 91 | 46 | -3.0 (-6.1%) | -2.8 (-5.6%) | -0.3 (-0.7%) |
| 11-02 | 111 | 56 | -7.7 (-10.5%) | -6.4 (-8.6%) | -1.5 (-2.0%) |
| 11-07 | 87 | 39 | -4.6 (-10.7%) | -4.6 (-10.6%) | -1.6 (-3.7%) |
| 11-08 | 82 | 39 | -3.0 (-7.0%) | -2.8 (-6.6%) | -0.6 (-1.4%) |