# Peer review of "Impact of aerosol optics on vertical distribution of ozone in autumn over YRD"

_Atmospheric Chemistry and Physics, 2022_

## Referee Comment (RC1)

**Review comments to manuscript acp-2022-752**

**Impact of aerosol optics on vertical distribution of ozone**

**by Yan S. et al.**

**Review Comments**

The paper seeks to quantify the impact of aerosols on ozone formation processes and vertical profiles in the boundary layer (BL) by combining direct observations from a ground-based site in Nanjing,China with a suite of WRF-Chem simulations. Specifically, three model runs are designed and performed to represent different aerosol mixing states (internal, core-shell and external mixing) and to quantify how ozone vertical profiles and photolysis rates are impacted with respect to a control run where no aerosol-radiation feedbacks are enabled. The key finding is that on the analyzed days, aerosols decrease ozone concentrations within the whole BL (although the signal is more pronounced at lower altitudes) during daytime with some sensitivity shown as a function of the mixing state.

Although the topic is of interest for Atmospheric Chemistry and Physics, the study is limited in the generalizability and robustness of its results, as well as in the model setup. Major concerns are listed in the general comments below.

**General comments:**

- A wider literature review needs to be incorporated to better provide background information on modeling features of aerosol mixing states and on prior studies on the topic, particularly in different parts of the world. This will provide a more complete statement of the problem, current uncertainties and gaps and further contextualize the presented results.

- A clear definition of the mixing states is missing and particularly how WRF-Chem treats them and what existing modeling limitations are. A discussion on the role of aerosol composition vs physical properties should be included. Please refer to the relevant literature including but not limited to the following:

  *Riemer, N., Ault, A. P., West, M., Craig, R. L., & Curtis, J. H. (2019). Aerosol mixing state: Measurements, modeling, and impacts. Reviews of Geophysics, 57, 187– 249. https://doi.org/10.1029/2018RG000615*

- Generalizability of the results. The WRF-Chem simulations are performed over a period of 4 days, while most of the presented analyses focus on a single day. Given the authors apply WRF-Chem in a configuration with a very small one-nested domain, the computational cost is very limited, so a justification on why the analysis is limited to such a few days should be provided, particularly in the context of the generalizability of the results. Robust statistics should be presented to make claims on physical and chemical mechanisms occurring in the atmosphere. Further, the authors should consider how different emission and meteorological conditions may play a role in impacting ozone

processes. For example, it would be relevant to analyze different ozone regimes by looking at a full year of simulations. At the least, simulations of a representative month for each season (or the season with most ozone formation) should be included.

**Specific Comments:**

- Line 51: as mentioned in the general comments, the three mixing states need to be defined and explained also in the context of the modeling tool adopted. Also, an explicit discussion on the role of aerosol composition should be included. Are measurements of aerosol composition available at that or nearby sites?
- Line 72: how much representative of the overall physics and chemistry of the atmosphere are the chosen days of November 2020 whose anthropogenic emissions may be still strongly impacted by COVID lockdowns?
- Line 73: How were the vertical profiles of meteorological and chemical species measured? At which heights?
- Line 82: a reference for WRF-Chem should be included
- Section 2.2 (Model configuration): More details and clarifications are needed. For example, the defined domains are unusually small (less than 100 x 100 grid cells) which raise concerns about the model's ability to develop proper meteorology and also chemical processes. Also, is 9 km resolution small enough to capture the spatial variability in ozone? Are there other ground-based sites in the region that would enable a more complete model evaluation (so at more than one point)? The authors mention the MEIC emissions are used. What is their temporal and spatial resolution? From the listed website it appears that the latest emission inventory available is for the year 2017. However, WRF-Chem was run over November 2020, a year that experienced significant changes in most anthropogenic emissions due to COVID lockdowns. How was this mismatch in emissions accounted for? Also, how often were the boundary conditions from ERA-5 and chemical species updated?
- Line 114: the quantified WRF-Chem skills should be put into the context of other literature studies (not necessarily in the same region) to verify if the model performance with the proposed setup is aligned with prior published work. Adding more sites to the evaluation and extending the simulation to more days/months will make the assessment of WRF-Chem performance more robust.
- Line 200: how is ADVC defined and computed?

---

## Author Comment (AC1)

**Response to Referee#1**

Dear Referee,

Thanks for giving us an opportunity to revise our manuscript (acp-2022-752). We appreciate your constructive comments and suggestions. We have studied them carefully and made revisions on the manuscript. These comments, suggestions and the corresponding replies are listed below.

Note that the title has been changed to "Impact of aerosol optics on vertical distribution of ozone in autumn over YRD" to clarify the study region and applicability.

The referee's comments are highlighted by gray. Followed by the comments are our responses. The texts led by **line number** are the current texts in manuscript, with some important revisions colored by red. The underlined blue texts, e.g., See the Response to Comment S6, means that the detailed information are provided in our response to the comment numbered with S6.

With regards,

Shuqi Yan, Bin Zhu*, and all co-authors.

**General comments:**

**G1**. A wider literature review needs to be incorporated to better provide background information on modeling features of aerosol mixing states and on prior studies on the topic, particularly in different parts of the world. This will provide a more complete statement of the problem, current uncertainties and gaps and further contextualize the presented results.

Thanks for this suggestion. We have addressed this comment by the following aspects.

**1)Basic concepts of aerosol mixing and roles of physical and chemical properties**

The influences of aerosol morphology, hygroscopicity, coating process and chemical composition on aerosol optics have been discussed in the Introduction (See the Response to Comment G2).

**2)Definitions of the three mixing states and model limitations**

The brief definitions of aerosol mixing states, model treatments on aerosol optics and the model limitations are included in Section 2.3 (See the Response to Comment G2).

**3)Model evaluations compared with previous works**

We add more sites and calculate the statistic metrics on meteorology, $PM_{2.5}$ and ozone. The statistical metrics are compared with previous studies (See the Response to Comment S6).

**G2**. A clear definition of the mixing states is missing and particularly how WRF-Chem treats them and what existing modeling limitations are. A discussion on the role of aerosol composition vs physical properties should be included. Please refer to the relevant literature including but not limited to the following:
*Riemer, N., Ault, A. P., West, M., Craig, R. L., & Curtis, J. H. (2019). Aerosol mixing state: Measurements, modeling, and impacts. Reviews of Geophysics, 57, 187– 249. https://doi.org/10.1029/2018RG000615*

Thanks for this suggestion. We have addressed this comment by the following aspects.

**1)The definition of mixing states, WRF-Chem treatments on mixing states and model limitations:**

We have added brief definitions of the mixing states and the model treatments on aerosol optics in Section 2.3. The treatment of mixing states, including aerosol species, aerosol number distribution, and formula of optical parameters of MOSAIC sectional approach are reasonably documented by previous papers (e.g., Fast et al., 2006; Grell et al., 2005), which have been adopted in this study and cited in the manuscript.

One major limitation of WRF-Chem model is basically discussed in Section 2.3. The real-world aerosol mixing state varies with emission, meteorology, chemical composition and other factors. The WRF-Chem, as well as other popular 3D models, is hard to trace and resolve the dynamic evolution and impact factors of aerosol mixing state at present state. The three mixing states in this study are idealized cases, which will inevitably cause the simulated aerosol optics deviating from observation.

**In Section 2.3**

In this work, the effect of aerosol optics on ozone profiles is addressed by its mixing states. We study three ideal types of mixing states: internal mixing, core-shell mixing and external mixing, which depend on the mixing behavior hypothesis of scattering and absorbing components. In internal mixing state, the relative fractions of chemical species in one particle are the same as that of the bulk aerosols. The complex refractive index (RI) of bulk aerosols is calculated by the volume-averaged RI of all aerosol species, and then it is passed to Mie optical module to calculate the required optical parameters (e.g., scattering coefficient, absorbing coefficient and single scattering albedo). The detailed formulas of aerosol optical parameters for MOSAIC sectional scheme are documented by previous works (e.g., Fast et al., 2006; Grell et al., 2005). In core-shell mixing, aerosol particles are hypothesized to be concentric spheres with BC as the core and non-BC aerosols as the coating shell (Riemer et al., 2019). The RI of the shell is the volume-averaged RI of non-BC aerosols, and the optics of core-shell mixed particles can also be treated by the Mie optical module (Ackerman & Toon, 1981). In external mixing state, each particle contains only one species with fixed optical characteristics. It is not included in the current WRF-Chem model, and the approximate treatment has been proposed by Gao et al. (2021b). In general, the Mie optical module separates BC aerosols from the bulk aerosols, and treats the optics of nonBC and BC aerosols individually.

To study the aerosol effect on ozone, four experiments are conducted (Table 2). The case "int" is the base experiment (the default option in WRF-Chem), in which the aerosols are internally mixed. The cases "csm" and "ext" are core-shell mixing and external mixing, respectively. The case "noARI" turns off aerosol-radiation feedback by setting aerosol optical depth as zero in radiation and photolysis modules. Therefore, the difference between noARI and three other experiments indicates the effect of aerosols in the corresponding mixing state.

One should note that the real-world aerosol mixing state varies with emission, meteorology, composition, and other factors. The dynamic evolution of aerosol mixing state and its influencing factors have not been addressed in most current 3D models (Matsui et al., 2013). This work addresses aerosol optics by the three ideal mixing states, which will inevitably cause the simulated aerosol optics deviating from observation.

**2)The role of aerosol composition vs physical properties:**

At monthly or annual scales, the aerosol chemical compositions in East China are dominated by SNA (sulfate, nitrate and ammonium) (commonly larger than 50%), followed by OM (organic matter) and BC (Tao et al., 2017; Yang et al., 2011). The scattering components, SNA and some OM, account for the majority of total aerosol concentration, and the absorbing components (mainly related to BC-contained aerosols) commonly account for less than 10% of total aerosol concentration (Tan et al., 2020, 2022).

We agree that aerosol physical properties (e.g., morphology, hygroscopicity) and chemical composition notably influence aerosol optical properties. The scattering components (e.g., SNA) generally contribute dominantly to aerosol extinctions. The contribution of SNA to total aerosol scattering coefficient can reach up to 60% (Tian et al., 2015). Under high humidity conditions, the hygroscopic growth of SNA can further enhance its extinction by 2~3 times (Zeng et al., 2019). Although BC takes a small proportion in aerosol mass concentration, the light absorption of BC contributes more than 70% to aerosol absorption coefficient (Yang et al., 2008), and the mass

absorbing efficiency of BC is comparable to the mass scattering efficiency of PM$_{2.5}$ (Tao et al., 2017). Additionally, the morphology of BC and BC-related coating process can change aerosol mixing state and optical properties (Bond et al., 2006; Liu et al., 2019). The BC light absorption can be amplified by a factor of 50~200% due to coatings (Cappa et al., 2012; Jacobson, 2001; Liu et al., 2017).

Aerosol mixing states also have significant effects on optical properties. The relative importance of aerosol mixing state and chemical composition on aerosol optics can be inferred from Zeng et al. (2019). Supposing aerosol is composed of BC and sulfate, where BC mass fraction is 5%. Aerosol exerts negative radiative forcing (RF) at the near surface in all mixing states. When the mixing state is changed from external to core-shell mixing (sulfate coating on BC), the decrease in RF (ΔRF) is 7.5W/m$^2$. The ΔRF becomes 7.9W/m$^2$ if sulfate is completely replaced with organic matter. The variation in ΔRF induced by coating material change is slighter than that by mixing state under various RH conditions. Curci et al. (2015) quantified the sensitivity of aerosol optical properties to aerosol mixing state, chemical composition and other parameters. Aerosol mixing state is found to be the dominant factor introducing uncertainties, explaining 30~35% of the uncertainty in AOD and single scattering albedo. Therefore, we infer that the effect of mixing state on aerosol properties could be more important than the effect of chemical composition.

**In Introduction**

The effect of aerosols on BL is related to aerosol optics, which are determined by aerosol morphology (Liu et al., 2019), hygroscopicity (Zeng et al., 2019), coating process (Bond et al., 2006) and chemical composition. The aerosol chemical composition in East China is dominated by SNA (sulfate, nitrate and ammonium) (larger than 50%), followed by organic matter and BC (3~8%) (Yang et al., 2011; Tan et al., 2020, 2022). The contribution of SNA to total aerosol scattering coefficient can reach up to 60% (Tian et al., 2015), and BC accounts for more than 70% of total aerosol absorbing coefficient (Yang et al., 2009). Furthermore, aerosol optics are strongly affected by aerosol mixing states. Since the real-world mixing state is highly variable and hard to be explicitly resolved (Riemer & West, 2013), three typical mixing states are generally hypothesized by previous works: internal mixing, core-shell mixing and external mixing. The mixing state is largely affected by the mixing behavior of BC with other aerosol species. The freshly emitted BC is commonly externally mixed with other species, but it will become more internally mixed due to coating process (Riemer et al., 2019). The BC light absorption can be amplified by a factor of 50~200% after being coated with scattering aerosols (Cappa et al., 2012; Jacobson, 2001; Liu et al., 2017). ~~The mixing behaviour hypothesis of aerosol scattering and absorbing components yields three major mixing states: internal mixing, core-shell mixing and external mixing. In internal and core-shell mixing, BC absorption can be enhanced by 50~100% (Bond et al., 2006; Jacobson, 2001). In external mixing, the absorption ability is weaker but scattering ability is stronger (Zeng et al., 2019).~~ Accordingly, aerosol mixing state alters aerosol optical properties and affects its interactions with BL and photolysis. Gao et al. (2021b) found that aerosols result in smaller boundary layer height (PBLH) reduction in external mixing (11.6 m) than in core-shell mixing (24 m), consequently leading to different changes in photolysis rates and ozone concentration.

*Summary*: The discussions of mixing state definition, roles of aerosol composition vs physical properties are included now. In this work, the hypothesis of three ideal mixing states will inevitably cause the simulated optics deviating from observation. However, the major chemical compositions in the atmospheric aerosols (e.g., SNA, OM, BC) have been included in the model, and the effects of coating and hygroscopicity are also considered by optical modules. The effect of chemical composition on aerosol optics has been reasonably addressed in the three ideal mixing states. In future, the role of aerosol composition on aerosol optics should be further addressed.

**G3**. Generalizability of the results. The WRF-Chem simulations are performed over a period of 4 days, while most of the presented analyses focus on a single day. Given the authors apply WRF-Chem in a configuration with a very small one-nested domain, the computational cost is very limited, so a justification on why the analysis is limited to such a few days should be provided, particularly in the context of the generalizability of the results. Robust statistics should be presented to make claims on physical and chemical mechanisms occurring in the atmosphere. Further, the authors should consider how different emission and meteorological conditions may play a role in impacting ozone processes. For example, it would be relevant to analyze different ozone regimes by looking at a full year of simulations. At the least, simulations of a representative month for each season (or the season with most ozone formation) should be included.

Thanks for this suggestion. In general, we have done the additional works: 1) Extending the simulation for one month (15 Oct to 15 Nov). 2) Finding that domain size and domain resolution have limited influence on ozone simulations. 3) Adding more sites and providing statistic metrics on meteorology, $PM_{2.5}$ and ozone. It can reflect the effect of different aerosol levels and aerosol-meteorology feedback under different emissions. 4) Finding that aerosols consistently decrease ozone concentration in polluted and clean days, and the effect is stronger in polluted days as expected. The details can be referred to the Response to Comment S5, S6.

We agree that ozone regime and ozone chemistry are highly variable among different seasons and meteorological conditions. In this study, we focus on the ozone characteristic in autumn season, considering the seasonal synoptic situation over the Yangtze River Delta Region. In spring and summer, the weather conditions vary significantly and precipitation events frequently occur. In winter, the solar radiation is relatively weak, so the ozone concentration is commonly not high. In autumn, the Yangtze River Delta Region is dominated by calm weather conditions. The weak winds and less cloudy weather conditions are better for studying the effect of aerosols on ozone. Therefore, we conduct our field observations and model simulations in a representative month of autumn season (actually from 15 Oct to 15 Nov). The aerosol-ozone relationships are applicable to only autumn season.

**Specific Comments:**

**S1**. Line 51: as mentioned in the general comments, the three mixing states need to be defined and explained also in the context of the modeling tool adopted. Also, an explicit discussion on the role of aerosol composition should be included. Are measurements of aerosol composition available at that or nearby sites?

Thanks for this suggestion. We provide the definition of three aerosol mixing states and discuss the importance of aerosol chemical composition in the Response to Comment G2.

The aerosol chemical composition measurements (e.g., SNA) are not available at nearby sites in the autumn of 2020. We have investigated some previous observations conducted in Nanjing in recent years. Generally, SNA are the dominant component in $PM_{2.5}$, with the proportion of about 58~81% (Guo et al., 2019; Liu et al., 2019; Liu et al., 2021; Sun et al., 2020; Wang et al., 2016; Wang et al., 2019; Yu et al., 2019). The BC commonly takes a minor part in $PM_{2.5}$. Tan et al. (2020, 2022) and Shen et al. (2021) reveal that the ratio of $BC/PM_{2.5}$ is approximately in the range of 3~8% in autumn season. Our simulations indicate that the ratio of BC to $PM_{2.5}$ is 1.9~5.6%, and the ratio of $SNA/PM_{2.5}$ is 55~75%, which are generally consistent with previous observations. Therefore, the simulation in this study can reasonably reproduce the observed aerosol composition and address the effect of composition on optics. Additionally, in the Response to Comment G2, we have suggested that the effect of aerosol composition on aerosol optics is reasonably addressed in the three ideal mixing states.

**S2**. Line 72: how much representative of the overall physics and chemistry of the atmosphere are the chosen days of November 2020 whose anthropogenic emissions may be still strongly impacted by COVID lockdowns?

Thanks for this suggestion. We have collected the MEIC emissions in 2019 and 2020 during the manuscript revision. Table X1 presents the reductions of anthropogenic emissions in 2020 (during the pandemic) with respect to 2019 (before the pandemic). In March 2020 when China is undergoing strict lockdowns, the $PM_{2.5}$ and $NO_X$ emissions are reduced by 10~15% compared with 2019. In November, the pandemic is effectively controlled in China. $PM_{2.5}$ and $NO_X$ emissions are reduced by at most 2.1%. It indicates that during the study period, COVID lockdowns have limited impacts on the emissions over China. We believe that the physics and chemistry of the atmosphere are still representative. **The MEIC emission in current manuscript has been changed from 2016 based to 2020 based now**.

Table X1. Relative reductions of $PM_{2.5}$ and $NO_X$ in 2020 compared with 2019. The comparisons are performed in March (strict lockdowns in China) and November (lockdowns in only a few cities). The units are $10^4$Mg for $PM_{2.5}$ and $10^4$ Mmol for $NO_X$.

| | East China (110~125°E, 25~45°N) | | Whole China | |
|---|---|---|---|---|
| | $PM_{2.5}$ | $NO_X$ | $PM_{2.5}$ | $NO_X$ |
| 201903 | 31.9 | 114.5 | 54.1 | 178.9 |
| 202003 | 28.1 | 98.8 | 48.7 | 155.7 |
| reduction | 11.9% | 13.7% | 10.0% | 13.0% |
| 201911 | 33.0 | 122.2 | 56.9 | 194.2 |
| 202011 | 32.3 | 122.0 | 56.1 | 191.5 |
| reduction | 2.1% | 0.1% | 1.4% | 1.4% |

**S3**. Line 73: How were the vertical profiles of meteorological and chemical species measured? At which heights?

The meteorological elements are measured by XLS-II tethered balloon system with a sounding balloon. The data are sampled every second until it loses signal. The air pollutants sensors are mounted on UAV platform. The UAV climbs vertically from the ground to about 1 km with a speed of 2m/s, and it descends along the same path with the same speed. Meteorology and air pollutants vertical data are averaged to 50 m intervals. The introduction of observation instruments of $PM_{2.5}$, BC and ozone can be referred to Shi et al. (2020, 2021). These texts are added into Section 2.1.

> The initial and boundary fields of meteorology are provided by ERA5 0.25°×0.25° reanalysis data. The chemical initial and boundary fields are provided by WACCM. They are all updated every 6 hours.

**S6**. Line 114: the quantified WRF-Chem skills should be put into the context of other literature studies (not necessarily in the same region) to verify if the model performance with the proposed setup is aligned with prior published work. Adding more sites to the evaluation and extending the simulation to more days/months will make the assessment of WRF- Chem performance more robust.

Thanks for this suggestion. We have extended the simulation to one month and evaluated model performance at four additional sites around Nanjing (Figure 2). The statistical metrics of temperature, wind speed, wind direction, PM$_{2.5}$ and ozone temporal variations are provided in Table 3 in the main text. Although a few metrics slightly exceeds its benchmark values, most of them are in acceptable ranges. The statistical metrics of PM$_{2.5}$ and ozone are consistent with previous works (e.g., Chen et al., 2022; Hu et al., 2016; Singh et al., 2012; Zhang et al., 2014). It increases the robustness of our simulation results.

Additionally, we compare the aerosol effect on ozone during some representative polluted and clean days. It is found that BL ozone is generally reduced by aerosol effect, and the reduction is more significant in polluted conditions.

[revised manuscript text omitted]

**S7**. Line 200: how is ADVC defined and computed?

The dynamic modules of WRF-Chem model can diagnose all the physical and chemical processes contributing to ozone variation. The ozone variation is affected by the following processes:

$$\frac{\partial O3}{\partial t} = \underbrace{-\left(u\frac{\partial}{\partial x}+v\frac{\partial}{\partial y}+w\frac{\partial}{\partial z}\right)O3}_{\text{Advc}} + \left(\frac{\partial O3}{\partial t}\right)_{\text{Vmix}} + \left(\frac{\partial O3}{\partial t}\right)_{\text{Chem}}$$

The first term is ADVC, i.e., the ozone tendency (ppb/h) caused by horizontal and vertical advections.

---

## Author Comment (AC2)

**Response to Referee#2**

Dear Referee,

Thanks for giving us an opportunity to revise our manuscript (acp-2022-752). We appreciate your constructive comments and suggestions. We have studied them carefully and made revisions on the manuscript. These comments, suggestions and the corresponding replies are listed below.

Note that the title has been changed to "Impact of aerosol optics on vertical distribution of ozone in autumn over YRD" to clarify the study region and applicability.

The referee's comments are highlighted by gray. Followed by the comments are our responses. The texts led by **line number** are the current texts in manuscript, with some important revisions colored by red.

With regards,

Shuqi Yan, Bin Zhu*, and all co-authors.

**General comments:**

More discussions are needed to clarify the meaning and limitations of this research. This is important for other researchers to consider the applicability of this study. Observations from the field campaign can be included to support the results concluded from model simulations. In addition, I suggest modifying the titles of section 3.2, 3.3 and 3.4 to convey the main topic of each section more clearly.

Thanks for this suggestion. We have addressed this comment by the following aspects:

1) Limitations of this research

The original simulation period is rather short (just several days). We have extended the simulation period to be one month (15 Oct to 15 Nov). We evaluate the model performance in the whole month, revealing that the model can reasonably capture the variation of temperature, wind, $PM_{2.5}$ and ozone (Section 3.1). We compare the aerosol effect on ozone under different pollution conditions, finding that aerosols cause more ozone reduction in polluted conditions than in clean conditions (Section 4).

This study is only applicable to autumn season. In spring and summer, the weather systems over Yangtze River Delta Region vary significantly and precipitation events frequently occur. In winter, the solar radiation is relatively weak, so the ozone concentration is commonly not high. Therefore, we conduct our field observations mostly in autumn season.

The WRF-Chem model has limitations in describing aerosol mixing state, which has been stated in the current manuscript (Section 2.3).

[revised manuscript text omitted]

2)Can observations from the field campaign support the results of model simulations?

A prior work by Shi et al. (2022) studies the effect of aerosols on photolysis and ozone profiles by observations from field campaign. It is found that aerosols inhibit ozone production in the lower BL and enhance photolysis and ozone production at upper BL. The observation data in this work is the subset of Shi et al. (2022).

**References**
Shi, S., Zhu, B., Tang, G., Liu, C., An, J., Liu, D., Xu, J., Xu, H., Liao, H., & Zhang, Y.: Observational evidence of aerosol radiation modifying photochemical ozone profiles in the lower troposphere, Geophys. Res. Lett., 49, e2022GL099274, https://doi.org/10.1029/2022GL099274, 2022.

3)The titles of Section 3.2~3.4 should be modified.

It has been modified to convey the exact meanings:
Section 3.2:  Impact of aerosols on BL and NO$_X$
Section 3.3:  Impact of aerosols on photolysis
Section 3.4:  Impact of aerosols on ozone profile

**Specific comments:**

S1. Line 76-78: "We mainly use the data from 2 to 5 November to study the effect of aerosols on ozone, and detailly investigate the physical and chemical mechanisms in the pollution stage on 2 November". I can not find related results in the manuscript.

In the current version, the simulation period has been extended from just a few days (2 to 5 November) to a month (15 October to 15 November). We have clearly stated when to use the whole simulation period and

when to use the single day of 2 November (in the leading text of Section 3).

**In Section 3**

It is an obvious pollution stage on 2 November 2020. The model evaluation on profiles (Section 3.1) and the mechanism of aerosols affecting ozone variation at the Nanjing site (Sections 3.2 to 3.4) are presented during that day. The model evaluation on time series (Section 3.1) and the aerosol effect under different pollution conditions (Section 4) are presented during the simulation period (15 October to 15 November).
* * *
S2. Line 88: Which year's emission inventory was used in this study?

The original year is 2016. In the current version, we have acquired new inventories from MEIC Group, so the base year is changed to be 2020, the exact year of the simulation period.
* * *
S3. Section 3.2: I'm confused about the content in this section. Why did you just describe the changes of NOx affected by aerosol-BL interactions instead of Ozone and PM2.5.

Thanks for this suggestion. We agree that the section titles did not convey the exact meanings. The section titles have been changed.

Section 3.2:  Impact of aerosols on BL and $NO_X$
Section 3.3:  Impact of aerosols on photolysis
Section 3.4:  Impact of aerosols on ozone profile
* * *
S4. Line 161-162: Does ozone here mean that in BL?

Yes. Our original focus is ozone within BL. We have stated it more clearly.

**In Section 3.4**

Figure 7 shows the ozone profile in various mixing states. We focus on the ozone within BL in the daytime. During 08:00~11:00, ……(the descriptions of BL ozone).
* * *
S5. Line 163: Should be "a strong positive gradient".

Thanks for this suggestion. This typo has been corrected.
* * *
S6. Section 3.4: Discussions about the differences between three aerosol mixing states in process analysis are rare, I suggest adding some content to explain the differences described in the first paragraph.

Thanks for this suggestion. We have added more discussions about ozone vertical variation, and removed some contents about the differences between three aerosol mixing states in process analysis.

**In Section 3.4**

Figure 7 shows the ozone profile in various mixing states. We focus on the ozone within BL in the daytime. During 08:00~11:00, the BL is in increasing stage, and ozone increases with height within BL. The average changes in ozone under internal, core-shell and external mixing are -9.7ppb (-15.8%), -8.5ppb (-13.8%) and -3.3ppb (-5.4%), respectively. As BL develops during 11:00~17:00, ozone shows a strong positive gradient near the surface, uniform distribution above the surface and negative gradient at upper BL. The average change in ozone under internal, core-shell and external mixing is -7.3ppb (-9.3%), -5.9ppb (-7.5%) and -1.0ppb (-1.2%), respectively. During the daytime (08:00~17:00), ozone reduction is larger in internal (10.5%) and core-shell mixing states (8.6%) and the smallest in external mixing state (2.0%). The reduction (about 3~13%) is the largest at near surface, which is due to that the $NO_X$ accumulation and photolysis inhibition are more profound at near surface. Other studies also reveal that ozone reductions caused by aerosols are approximately in the range of 10~20% (e.g., Gao et al., 2020; Qu et al., 2021; Yang et al., 2022). Above surface where the layer is more well-mixed, ozone reduction is relatively weaker. It can be inferred that diurnal ozone concentration is generally reduced in all mixing states and at all heights within BL. The reduction is the smallest in external mixing state. It could be because the enhanced NO titration effect associated with NOx accumulation is weaker in external mixing than in other mixing states

(Figure 4c). Also, externally mixed aerosols lead to less photolysis suppression in the lower level and larger photolysis enhancement in the upper level (Figure 6a and b), which will partly counteract the reduction in ozone concentration

**In Section 3.4**

~~Table 4 quantitatively describes the respective contributions of three processes to ozone variation during 11:00~17:00. From near surface to lower BL (0~300m), the positive VMIX contribution is stronger than the negative CHEM contribution, and the role of ADVC can be ignored. At lower to middle BL (300~800m), the promoting effect of VMIX on ozone weakens, and instead, the negative contribution of CHEM turns to positive and becomes the dominant influencing factor. At the upper BL (800~1500m), VMIX plays the dominant role due to the increasing ozone entrainment at upper BL (Figure 8b-d). The relative contributions of the three processes are generally consistent in all mixing states.~~

| |  |  |  |
|---|---|---|---|
| ~~H: 0~300m~~ | | | |
|  |  |  |  |
|  |  |  |  |
|  |  |  |  |
| ~~H: 300~800m~~ | | | |
|  |  |  |  |
|  |  |  |  |
|  |  |  |  |
| ~~H: 800~1500m~~ | | | |
|  |  |  |  |
|  |  |  |  |
|  |  |  |  |